

# Excitons under strain: Light absorption and emission in strained hexagonal boron nitride

Pierre Lechifflart[1⋆], Fulvio Paleari[2,3] and Claudio Attaccalite[1,2,4]

**1** CNRS/Aix-Marseille Université, Centre Interdisciplinaire de Nanoscience de Marseille
UMR 7325 Campus de Luminy, 13288 Marseille cedex 9, France
**2** CNR-ISM, Division of Ultrafast Processes in Materials (FLASHit), Area della
Ricerca di Roma 1, Via Salaria Km 29.3, I-00016 Monterotondo, Scalo, Italy
**3** CNR-NANO, Via Campi 213a, 41125 Modena, Italy
**4** European Theoretical Spectroscopy Facilities (ETSF)

⋆ lechifflart@univ-amu.fr

## Abstract

Hexagonal boron nitride is an indirect band gap material with a strong luminescence in the ultraviolet. This luminescence originates from bound excitons recombination assisted by different phonon modes. The coupling between excitons and phonons is so strong that the resulting light emission is as efficient as the one of direct band gap materials. In this manuscript we investigate how uniaxial strain modifies the electronic and optical properties of this material, and in particular how it affects the exciton-phonon coupling. Using a formulation of this coupling based on finite-difference displacements, recently developed by some of us, we investigate how phonon-assisted transitions change under strain. Our results open the way to the study of phonon-assisted luminescence in strained materials from first principles. Our findings are important both for experiments that directly probe *h*-BN under strain or for those in which it is used as substrate for other 2D material with a lattice mismatch.



# 1 Introduction

Hexagonal boron nitride (*h*-BN ) is a large band gap material, that has recently attracted attention from the scientific community as a very efficient light emitter in the ultraviolet [1] with an internal quantum yield of $\simeq 45\%$ [2]. This strong luminescence has been attributed to the recombination of an indirect exciton [3] assisted by atomic vibrations [4].

Beyond the bulk *h*-BN , its 2D counterpart also has many applications as a substrate for low dimensional devices, due to its large band gap and the low lattice mismatch with other 2D materials [5].

Strain can be used to modulate the band structure and engineer the material properties [6]. In particular it is possible to modify the position of the conduction band (CB) minima and valence band (VB) maxima thanks to strain, even leading to a transition from direct to indirect character of the band gaps. This has been shown for bulk Ge [7,8], and more recently in low dimensional transition metal dichalcogenides [9–11]. In the case of *h*-BN , different experiments were performed to study the effect of an hydrostatic pressure on the structural, vibrational [12,13] and optical properties [14]. The effect of strain on phonons and Grüneisen parameters was investigated experimentally in exfoliated *h*-BN with various thicknesses [15], while theoretically biaxial tensile strain was considered for the mono- and bilayer *h*-BN [16,17] as well as its role on hexagonal boron nitride quantum emitters [18]. But little is known about the effect of strain on the optical properties of bulk *h*-BN which are the simplest mean to characterize this material. In this manuscript we extend these works to investigate the effect of uniaxial strain on bulk *h*-BN , including also its effect on light absorption and emission.

This paper is organized into the following sections: we first detail our calculations including a discussion on strained cell construction. Next, phonon calculations are presented. Then we discuss how strain modifies electronic properties, and finally we investigate optical absorption and emission.

# 2 Computational details

In order to study the effect of uniaxial strain on *h*-BN , we built the strained simulation cell using a two-step procedure. First we construct an orthorhombic cell with 8 atoms at equilibrium, as shown in Fig. 1. This supercell has the advantage of having orthogonal lattice vectors, so that strain can be applied along a unique lattice vector, letting the other two free to relax.

We applied different strains along the x-direction, the one parallel to the B-N bond, ranging from −2.5% to +2.5% of the equilibrium cell vector. Then we allowed the cell vectors and atomic positions to relax only in the other two orthogonal directions, keeping the arbi-

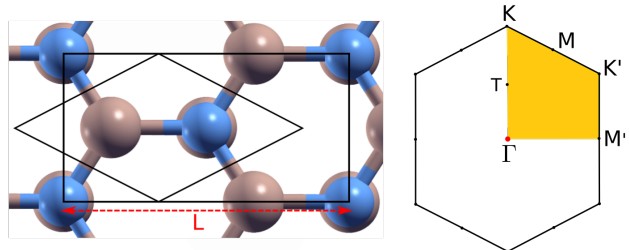

Figure 1: [Color online] Structure of strained hexagonal Boron Nitride along the B-N bonding, obtained by stretching the equilibrium orthorhombic cell. Pseudo-hexagonal cell is shown on the left, and the corresponding Brillouin zone is shown on the right.

trarily strained length fixed. The relaxation is done using Density Functional Theory and a damped molecular dynamics algorithm as implemented in the QuantumEspresso code [19], with norm-conserving pseudo-potentials in the Local Density Approximation (LDA), a kinetic energy cutoff of 120 Ry and an equivalent Monkhorst-Pack grid of $18 \times 18 \times 6$ $k$-points. The forces acting on the cell and atoms were converged to be lower than $10^{-6}$ a.u.

Once the strained orthorhombic cells were relaxed, we have constructed strained pseudo-hexagonal cells (see Fig. 1) in order to proceed with the electronic and optical calculations.

On these pseudo-hexagonal 2-atom cells we performed phonon calculations using Density Functional Perturbation Theory [19], with **q**-points and **k**-sampling respectively of $6 \times 6 \times 2$ and $18 \times 18 \times 6$, in order to verify the stability of our structure and the effect of strain on phonon modes.

The quasi-particle band structure was obtained within the $G_0W_0$ approximation, using again a $18 \times 18 \times 6$ k-points sampling, with 210 bands plus a terminator [20] for G and W in order to speed up convergence. We used a cutoff of 7 Ha for the dielectric constant that was calculated within the plasmon-pole approximation. Excitons and optical absorption were studied solving the Bethe-Salpeter equation [21] using 4 valence and 4 conduction bands, as implemented in the Yambo code [22], using the same k-points grid as for the $G_0W_0$ calculations.

Luminescence was calculated following the approach described in Ref. [23]. We searched for the minima of the indirect gap within the independent particle approximation and we used the corresponding **q**-vectors to construct a supercell that map these points at Γ. We displaced atoms along all possible phonon modes having a periodicity commensurate with the different **q**-points of the indirect gap minima, and calculated the derivatives of the excitonic optical matrix elements.[1] With these ingredients plus the phonon frequencies we reconstructed the spectra using the van Roosbroeck-Shockley (RS) relation [23]. More details and approximations on this kind of calculations are discussed in the next section.

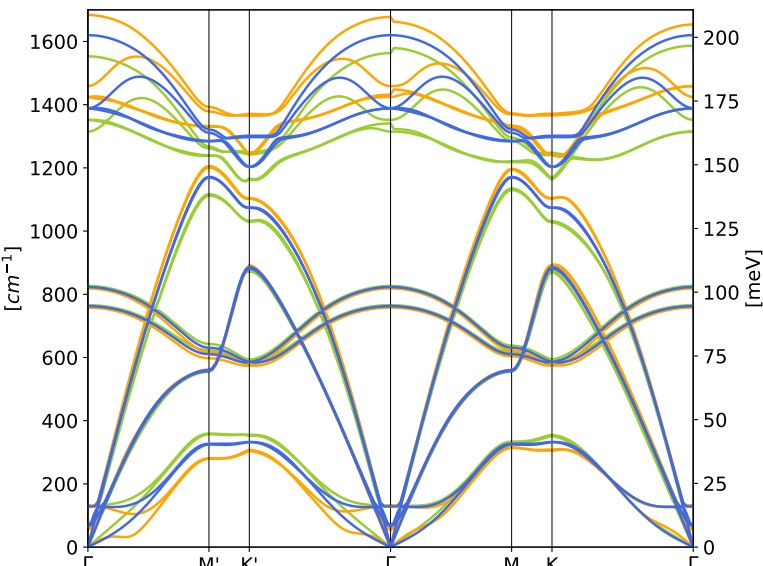

Figure 2: [Color online] Phonon band structure versus uniaxial strain. Blue lines are at equilibrium, green lines at 2.5% stretch and orange lines at 2.5% compression.

---

[1]In the luminescence calculations we used a smaller k-point sampling, $12 \times 12 \times 4$ and a scissor operator to speed up calculations in the supercells, similar to Refs. [4, 23]. These parameters are sufficient to describe the lowest exciton that is responsible for the luminescence.

## 3 Results

We start from the LDA equilibrium structure of h-BN, with a lattice parameter of $a = 2.45$Å and interlayer distance of $c = 3.22$Å. For the different applied strains both the new cell and the atomic positions are relaxed minimizing the enthalpy. We found that both lattice vectors perpendicular to the x-direction vary linearly with the applied strain. For each new cell we calculated vibrational frequencies and electronic properties. In Fig. 2 we report the phonon band structures for the equilibrium and the two maximum/minimum strains studied in this work.

In strained $h$-BN the 120° rotation symmetry is not present anymore and this makes the $\mathbf{M}, \mathbf{K}$ points not equivalent to their $\mathbf{M}', \mathbf{K}'$ counterparts in the pseudohexagonal cell. For this reason, we report phonon modes along the path depicted in Fig. 1. We found a large shift and split of the $E_{2g}$ mode in agreement with Raman measurements and previous calculations [15, 24]. The shift of the phonon frequencies depends on the strain and the phonon mode: under compression, the acoustic modes have their energies lowered; under stretch, the acoustic modes have their energies increased. The shift is reversed for the optical modes in the upper part of the dispersion spectra, that are increased under compression and decreased under stretch. We also found that the LA, TA and TO modes are less affected by strain compared to the other modes. This fact will be important later in the discussion of luminescence. Finally for large compression the system tends toward an instability, as it is possible to see from the softening of the ZA acoustic mode close to Γ. We did not investigate further this instability because it is beyond the strain range we were interested in.

The results of Fig. 2 guarantee us that the system is stable in the strain range we considered.

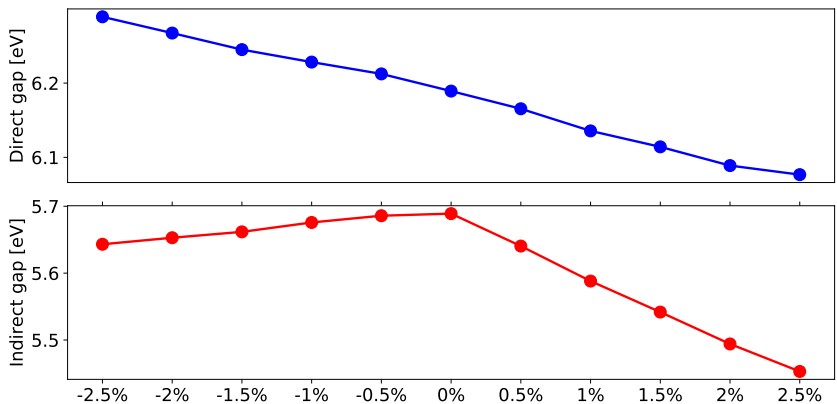

Figure 3: [Color online] Quasi-particle corrections to the direct and indirect gaps with respect to strain.

Now we move on to electronic and optical properties.

### 3.1 Electronic structure

In Fig. 3 we report the direct and indirect gaps at the $G_0W_0$ level versus strain. We found that the direct band gap changes linearly from negative to positive strain, while the indirect gap has its maximum at equilibrium and decreases both with compression or stretching. In both cases, the $G_0W_0$ correction is just a rigid shift of the DFT energies with respect to strain. A similar cell-independent shift was also found in the case of hydrostatic pressure [14]. The different behaviors of the direct/indirect gap originate from the different locations of the points in the Brillouin zone that form these gaps. At equilibrium, the indirect gap is located between $\mathbf{M}$

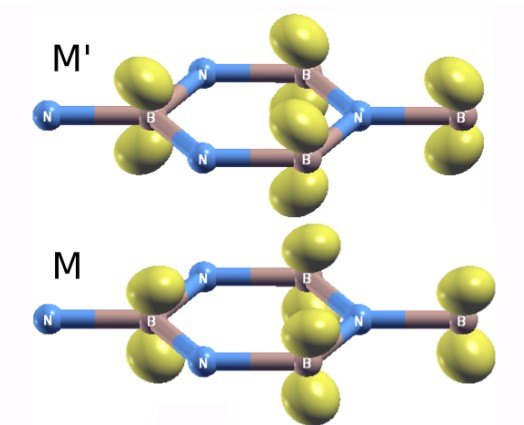

Figure 4: [Color online] The orbitals at **M** and **M**′. Notice that in principle there is another orbital along the third bonds, equivalent to the one at **M** not showed here.

and a point close to **K**, while the direct one is close to **K**. When strain is applied, however, the points **M**, **M**′ and **K**, **K**′ become inequivalent. In particular, under compression the minimum of the conduction band is located at point **M** and the maximum of the valence band is located at the point called $T_2$ close to **K**′. Under stretch, the conduction minimum is located at **M**′ and the valence maximum is located at $T_1$ close to **K**. The corresponding indirect gaps in the two cases are reported in Fig. 5. This happens because while the atomic-like orbitals at **M** and **M**′ are degenerate at equilibrium, when we apply a strain this degeneracy is lifted and the orbital along the compressed/stretched direction, at the **M**′ point, acquires an energy different from the ones at **M** that are oriented along the other two bonding directions, see Fig. 4. We found that compressing the system moves the **M**′ states up in energy, while stretching it decreases their energy with respect to the equilibrium geometry. Furthermore, the orbitals close to **M**/**M**′ are also affected by the interlayer interaction, while the states at **K** are protected for symmetry reasons [25]. The interlayer interaction changes with the distance between the planes which varies linearly with compression/stretching of the system, and this effect contributes to the in-plane changes [14].

Then, at **M**′, we see that both conduction and valence bands are shifted down under stretch, while they are shifted up under compression. The opposite trend appears instead for the states at point **M**, but the conduction band here shifts less. As a consequence, the indirect electronic gap is maximal in the equilibrium structure, and it decreases linearly with compression and stretching, while also changing position within the Brillouin zone, as outlined above and in Figs. 3 and 5.

### 3.2 Optical absorption

In bulk h-BN the lowest excitation is composed of four peaks degenerate two by two. The splitting between these pairs is due to the inter-layer interaction, the so-called Davydov splitting [26]. These two pairs of excitons are different for inversion symmetry, one is even and the other is odd. This makes the lowest pair dark in linear response and bright in two-photon absorption. [27] On the opposite the other pair is dark in two-photon absorption and forms a linear combination of bright and dark excitons in linear response, due to the rotation symmetry.

The effect of uniaxial strain on the direct exciton is double. First the exciton energy is shifted according to the change in direct gap, see panel (*a*) in Fig. 7. Second, due to 120-degree rotational symmetry breaking, the Davydov pairs are not degenerate anymore and we

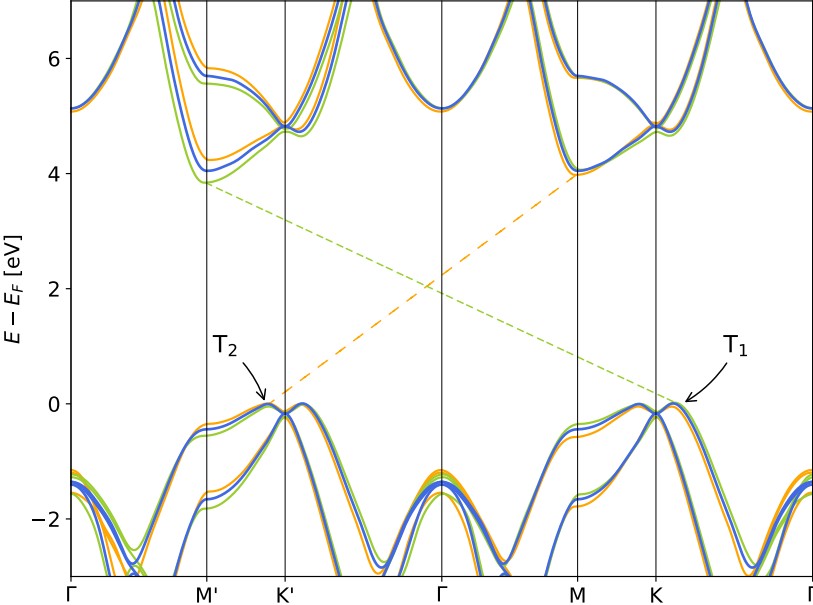

Figure 5: [Color online] Details of the electronic band structure under the maximum stretch and compression considered in the manuscript. Blue lines are at equilibrium, green lines at 2.5% stretch and orange lines at 2.5% compression. We report also the location of the new indirect gaps in the two cases. Notice that at equilibrium all indirect transitions between the different **K** and **M** points are equivalent.

observe a splitting of the excitonic energies of about 10 to 15 meV at the maximum compression/stretch. This splitting can be understood by plotting the exciton wavefunctions. As we can see from Fig. 6, excitons are split into one exciton localized along the strain direction and another along the other two bonds. Notice that under strain the lowest excitonic pair remains dark, because strain does not break the inversion symmetry. Instead the third and forth excitons are both bright with two distinct dipole matrix elements since they are not mixed anymore by the rotation symmetry (see panel (*b*) of Fig. 7).

We found that the shift of the main exciton peak in the strained structures is mainly due to the change in the gap, see Fig. 5. In fact the exciton binding energy remains almost a constant with respect to strain with a maximum change of about 5 meV. From the gap change we can estimate the strain gauge factor, defined as the spectral shift per % of uniaxial strain. For uniaxial straining direction along the B-N bonding we found a value of $\simeq 43$ meV/%, similar to the one found in transition metal dichalcogenides (typically between 30-60 meV/%) [28]. Notice that the strain gauge factor could have a strong dependence on the angle on which the strain is applied. [29]

Since *h*-BN is an indirect material, excitons at finite momentum will also contribute to its optical properties as it has been recently shown by C. Elias et al. [1]. We found that indirect excitons, i.e., electron-hole bound pairs having a finite momentum that connects the different maxima/minima of the valence and conduction bands, Fig 5, are shifted according to the change of the indirect gap, see panel (*c*) in Fig. 7 and Fig. 3. In fact also for the indirect excitons their binding energy changes very little with the strain. We expect these changes will be visible in reflectance experiments in addition to the change of the assisting phonon frequencies. More details on phonon-assisted transition are discussed in the next section on luminescence.

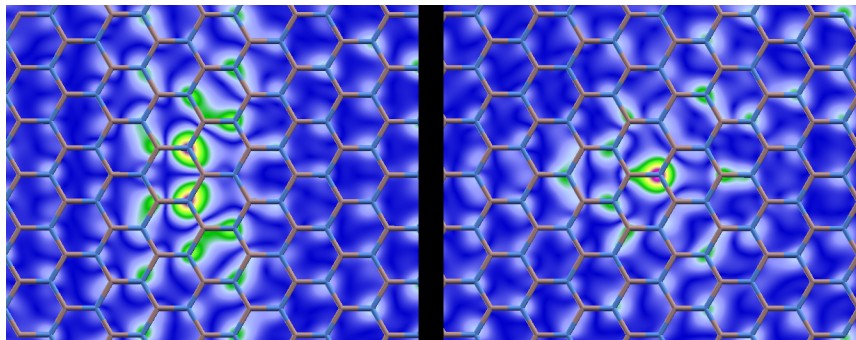

Figure 6: [Color online] Lowest excitonic wave-functions, generated by putting the hole on the nitrogen atom. The two wave-functions, one along the strained bond and the other one along the other two binds, are not degenerate anymore in the strained cell.

## 3.3 Luminescence

Luminescence in indirect materials can be studied by taking into account atomic vibrations that couple with indirect excitons. For $h$-BN this has been recently done by means of finite difference approaches [4,23] and by direct calculation of the exciton-phonon matrix elements [30]. The exciton-phonon matrix elements correspond to the derivatives of the excitonic dipole matrix elements with respect to the phonon modes [4]. In a finite difference approach one constructs the supercells containing the phonon modes responsible for the light emission, and displaces atoms along these phonon modes to calculate these derivatives.[2] Then luminescence can be calculated using the expression derived in Refs. [23,31]. This formula is a generalization of the van Roosbroeck-Shockley relation [32] to the excitonic case. The light emission reads:

$$R^{\mathrm{sp}}(\omega) = \sum_{\lambda,\overline{q}} \frac{\omega(\omega + 2\Omega_{\lambda\overline{q}})^2}{\pi^2 \hbar c^3} n_r(\omega) \sum_S \frac{\partial^2 |T^S|^2}{\partial R^2_{\lambda\overline{q}}}\bigg|_{\mathrm{eq}} \Im\left\{\frac{1}{\hbar\omega - (E^S - \Omega_{\lambda\overline{q}}) + i\eta}\right\} B\left(E^S_{\overline{q}}, T_{exc}\right), \quad (1)$$

where $n_r(\omega)$ is the refractive index, $B\left(E^S, T_{exc}\right)$ is the Bose occupation of excitons, $T_{exc}$ is the excitonic temperature, $\frac{\partial^2 |T^S|^2}{\partial R^2_{\lambda\overline{q}}}$ are the derivatives of the exciton dipole matrix elements for a given phonon mode, $\Omega_{\lambda\overline{q}}$ are the phonon frequencies and the sum on $\overline{q}$ is on the different supercells obtained from phonon modes responsible for the luminescence. The excitonic eigenvalues and eigenvectors used to construct the luminescence spectra are obtained by means of the solution of the so-called Bethe-Salpeter equation [21], where we use a scissor operator to simulate the quasi-particle correction of the single particle levels. This equation is solved in each supercell, for all possible displacements compatible with the chosen momentum $\overline{q}$ in such a way to calculate dipole second-order, finite-difference derivatives (more details can be found in Ref. [23]). In Eq. 1 we omitted the Bose factor for the phonon occupation that at low temperature can be approximated to one. For the Lorentzian broadening we use a linear model $\eta = \Gamma_0 + aT + bB(T)$, with the same parameters as Refs. [23,33], where $T$ is the lattice temperature and then we derive the excitonic temperature $T_{exc}$ using the results of Ref. [34]. Notice that with respect to the works of Ref. [23] we did a further approximation in the derivative of the dipole matrix elements. In general the dipole matrix elements depend on the two-particles Green's function $L$ and the screened interaction $W$, that are the ingredients used to build the

---

[2]These operations have been performed with the aid of yambopy, a python-based interface to Yambo and QuantumEspresso.

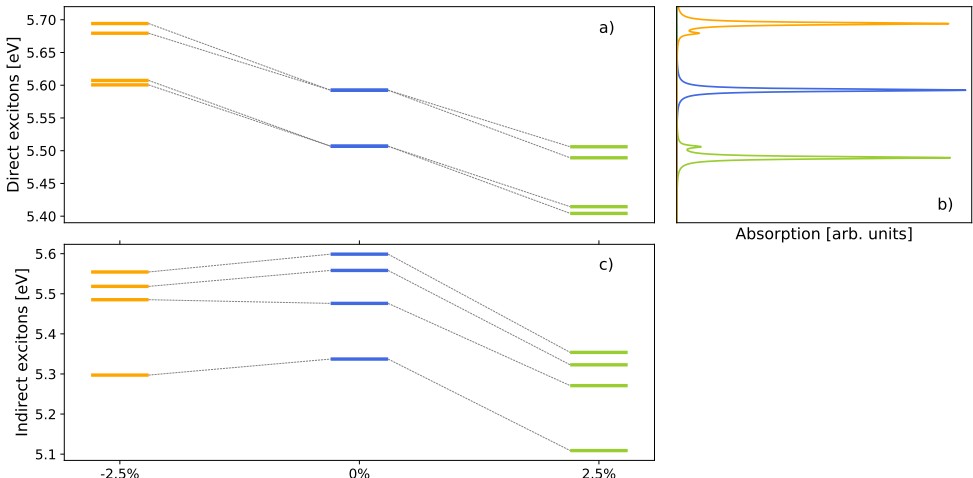

Figure 7: [Color online] In panel ($a$) we report the energy of the lowest direct excitons with respect to strain, and in panel ($b$) the corresponding absorption spectra. In panel ($c$) the lowest indirect excitons vs strain are reported. As in the previous Figures, blue lines are at equilibrium, green lines at 2.5% stretch and orange lines at 2.5% compression.

Bethe-Salpeter equation. In this work we approximate these derivatives as:

$$\frac{\partial^2 |T^S(W,L)|^2}{\partial R^2_{\lambda \bar{q}}} \simeq \frac{\partial^2 |T^S(W(R=R_{eq}),L)|^2}{\partial R^2_{\lambda \bar{q}}} \ . \tag{2}$$

In practice what we did is to calculate $W$ at equilibrium and use this interaction in the cell with the displaced atoms to build the Bethe-Salpeter equations. A similar approximation was already employed in the calculation of electron-phonon matrix elements [35], where it has been shown that keeping W fixed at equilibrium position has negligible effects. We verified that this approximation does not modify the final luminescence spectra.

The finite difference approach has the advantage of not requiring any special code for the calculations, but it is limited to simple systems and phonon modes that do not map on too large supercells. On the other hand, with finite differences it is not possible to take into account the full exciton and phonon dispersions in the luminescence. We will come back to this point later. For the bulk $h$-BN at equilibrium the **q**-vector responsible for the exciton recombination can be approximated with $\mathbf{q}=(\frac{1}{3},-\frac{1}{6},\mathbf{0})$ that is the q-vector that connects the **M** and **K** points in the Brillouin zone. The corresponding exciton with momentum $\mathbf{q}=\mathbf{K}-\mathbf{M}$ in $h$-BN is the so called $iX_1$ exciton.

In the strained case, things are more complicated. The presence of strain breaks the equivalence between **K** and **K′**, **M** and **M′** as shown in Fig. 5. Different indirect excitons with very close energies can now contribute to the luminescence. These excitons will scatter with phonons, themselves modified by the presence of strain. In particular since optical modes are more impacted by strain compared to acoustic ones, see Fig. 2, we can expect that the lowest energy peaks in luminescence will be further split in many sub-peaks. In order to verify this hypothesis we construct the different supercells corresponding to the transitions $\mathbf{M}-\mathbf{K}$, $\mathbf{M}-\mathbf{K'}$, $\mathbf{M'}-\mathbf{K}$, $\mathbf{M'}-\mathbf{K'}$ and sum contributions in order to obtain the final luminescence spectra reported in Fig. 8. For zero strain, we obtain the standard equilibrium luminescence spectra of $h$-BN see Fig. 8. At low strain the excitons originating from the different $\mathbf{M}^{(')}-\mathbf{K}^{(')}$ transitions have a

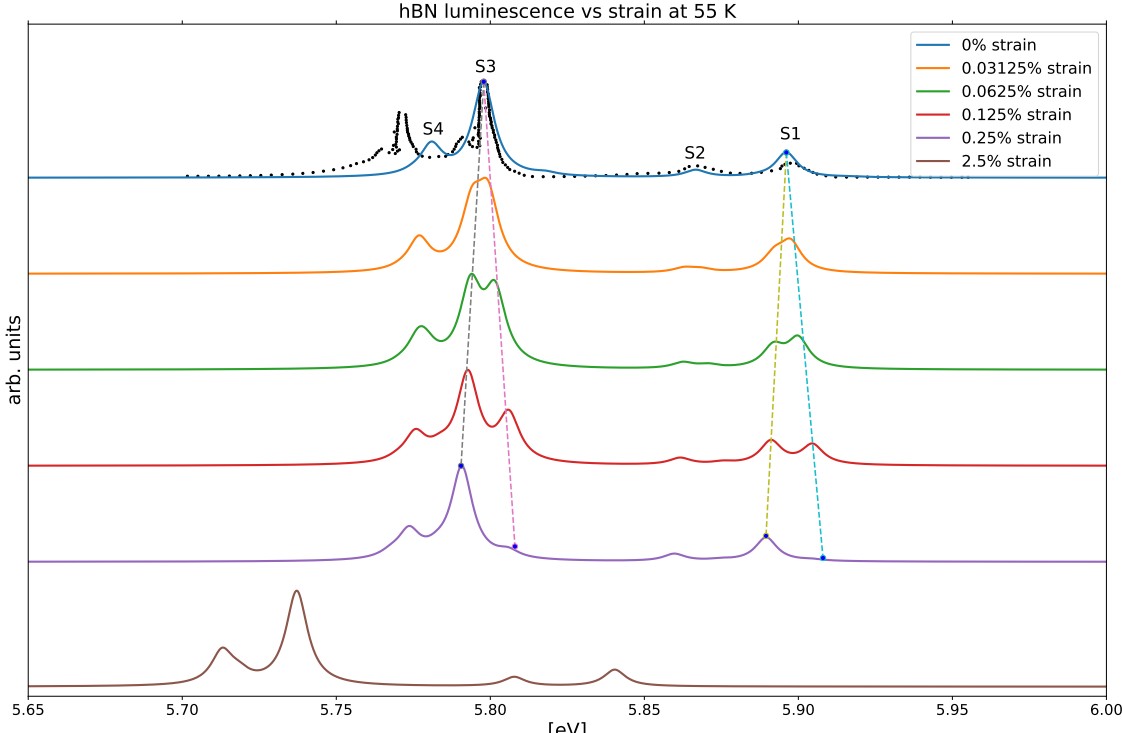

Figure 8: [Color online] Luminescence versus strain, for different compressions. On the top plot, experimental data is represented by the black dots from Ref. [36]. The spectra has been shifted to match the position of the $iX_1$ exciton at equilibrium, and compensate the difference between one-shot and self-consistent GW [37]. Dashed lines are just for the eye.

very close energy and therefore all of them contribute to the luminescence spectra. These excitons scatter with phonons modes which in turn have been modified by the presence of strain. This generates a splitting of the main peaks in the luminescence spectra, and a slight increase of the intensity of the peaks assisted by acoustic modes, see Fig. 8. At equilibrium we found that the ratio between the S3/S1 peak intensities is $\simeq 3.7$ while for small strain it decreases down to $\simeq 2.7$. This result goes in the direction of L. Schue et al. [38] measurements that found a reduction of this ratio up to $\simeq 1$ in their compressed structures.

But experiments and theory are still qualitatively far away, the reduction of the S3/S1 ratio we found is too small compared with measurements. This could be attributed to the lack of a fine sampling for the exciton and phonon dispersions [30], that is not possible with a finite difference approach or to the presence of surface effects [26] not included in the present calculations.

Finally for larger compression the energy differences between the different indirect excitons become larger and larger and only the lowest exciton contributes to the spectra while the higher energy peaks are suppressed by the Boltzmann factor in Eq. 1. The spectrum thus acquires the same shape as the equilibrium one but translated to lower energies due to the closure of the indirect gap, and the change in the phonon frequencies. In the stretching case, not reported in the manuscript, results are similar.

## 4 Conclusions

In this manuscript we studied electronic and optical properties of $h$-BN as a function of uniaxial strain along the BN bonding. We observe a splitting of the exciton at $\Gamma$ due to the breaking of the threefold rotational symmetry. The splitting could be measured in reflectivity experiments [1]. We also found that the direct and indirect excitons shift in a different way: the direct exciton energy variation is linear with strain, while the indirect exciton has its maximum energy at equilibrium. The strain gauge factor of the main exciton in $h$-BN has been found similar to the one in transition metal dichalcogenides. Then we investigated the luminescence using a finite difference approach. We found that at low strain, additional peaks are present in the spectra due to the breaking of the degeneracy between the different $\mathbf{K}$ and $\mathbf{M}$ points in the Brillouin zone. These additional peaks decrease the intensity ratio between the acoustic- and the optical-phonon assisted transitions, in agreement with recent measurements. For large strain we found that only one valley contributes to the luminescence spectra, and the spectra return to a shape similar to the equilibrium one but shifted at lower energies. This prediction could be verified by means of luminescence measurements in highly strained $h$-BN [24].

## Acknowledgments

The research leading to these results has received funding from the European Union Seventh Framework Program under grant agreement no. 696656 Graphene Core1 and no. 785219 Graphene Core2. C.A. and P.L. acknowledge A. Saul and K. Boukari for the management of the computer cluster *Rosa*. This publication is based upon work from COST Action TUMIEE CA17126, supported by COST (European Cooperation in Science and Technology). CA acknowledges funding through the MIUR PRIN (Grant No. 2020JZ5N9M). We acknowledge J. Barjon, E. Cannuccia, A. Loiseau for the useful discussions. We dedicate this manuscript to the memory of François Ducastelle.

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
