# Peer review of "Excitons under strain: light absorption and emission in strained hexagonal boron nitride"

_SciPost Physics, doi:SciPost Phys. 12, 145 (2022)_

## Round 2 · Referee Report · Anonymous (Referee 1) · 2022-2-4

Report

The work presented in "Excitons under strain: light absorption
and emission in strained hexagonal boron nitride" is an original
and sound study of the effects of strain in optical properties
of hBN. It lies within ab initio Green's functions theory, by solving
the Bethe-Salpeter equation to obtain the optical absorption.
The photoluminescence is studied coupling the BSE results
(in particular the indirect excitons) with phonon calculations.
This constitutes the state-of-the-art for a theoretical approach to absorption and photoluminescence.
The results are carefully analysed, but the paper's clarity will
profit from some adjustments and complementary information.

Requested changes

1) it is not clear what kind of calculations were done on the pseudo-hexagonal cell and how. Once one relaxes the orthorhombic cell, I guess it will be difficult to find a smaller (only 2-atoms) hexagonal cell to be repeated in the 2 space dimensions. So how does this work? For instance, has the absorption been evaluated on a 2-atoms cell or on the orthorhombic one?

2) computational details are reported for the phonons calculations, as well as for the DFT and GW calculations. However, for the main focus of the article, the exciton, little info is available. For example, how many k-points have been used (hBN has a slow convergence in the BZ sampling)? Especially because the authors claim to resolve a degeneracy within 10 meV.

3) the optical absorption spectra are commented, but not shown. At the very least the different curves should be shown, as a function of the strain/compression.

4) It would be nice to see the extra features that appear when the unilateral strain is applied. Why is it there, when it was dark at equilibrium? This kind of analysis should not be too difficult having the BSE results at disposal.

  • validity: high
  • significance: high
  • originality: top
  • clarity: good
  • formatting: -
  • grammar: good

Author:  Claudio Attaccalite  on 2022-03-11  [id 2282]

(in reply to Report 1 on 2022-02-04)
Category:
answer to question

We thank the referee for his/her comments and remarks.

>1) it is not clear what kind of calculations were done on the
>pseudo-hexagonal cell and how. Once one relaxes the orthorhombic
>cell, I guess it will be difficult to find a smaller (only 2-atoms)
>hexagonal cell to be repeated in the 2 space dimensions. So how
>does this work? For instance, has the absorption been evaluated
>on a 2-atoms cell or on the orthorhombic one?

As explained in the attached PDF, the two cells are equivalent.
The only reason to use an orthorhombic cell is that during the optimization run
it is more difficult to impose a constraint on one Cartesian direction while
optimizing the other two in a hexagonal cell, while it is automatically implemented
for orthorhombic cells in codes like QuantumEspresso.
We geometrically verified that the pseudo-hexagonal and orthorhombic cells
reconstruct the exact same crystal.
Then, except for cell optimization, all other calculations
were performed in the pseudo-hexagonal one: phonons, band structure, optics.
We added a sentence in the manuscript to clarify this point.

>2) computational details are reported for the phonons calculations,
>as well as for the DFT and GW calculations. However, for the main
>focus of the article, the exciton, little info is available.
>For example, how many k-points have been used (hBN has a slow
>convergence in the BZ sampling)? Especially because the authors
>claim to resolve a degeneracy within 10 meV.

We thank the referee for this question, we added all the computational details
for the BSE at equilibrium as well as for the calculation of the exciton-phonon coupling.
The listed parameters lead to converged results for the low-energy features of both absorption and luminescence spectra.

>3) the optical absorption spectra are commented, but not shown.
>At the very least the different curves should be shown, as a function
>of the strain/compression.

We modified Fig. 7 of the manuscript by adding the corresponding BSE absorption spectra
under strain along with the exciton energy levels.

>4) It would be nice to see the extra features that appear when the
>unilateral strain is applied. Why is it there, when it was dark at
>equilibrium? This kind of analysis should not be too difficult having
>the BSE results at disposal.

These features appear now in the absorption spectra added to Fig. 7, where we emphasised
the splitting of the two main doubly-degenerate excitons.
We also added a discussion in the text on the role of inversion symmetry that is at the origin
of the dark/bright excitonic pairs in h-BN and the 120 degree rotation that is broken by
the strain and induces two exciton peaks in the absorption spectra.
We also removed a previous mistake when we wrote that the dark exciton would acquire a small oscillator strength upon strain.
This is wrong since as long as inversion symmetry is not broken, this exciton - split by strain or not - is forbidden by selection rules. The two peaks in the absorption spectra under strain are due to exciton 3 and 4 that both are bright under strain due to the breaking of the 120 degree rotation.
We gratefully thank the referee for pointing this out.

Attachment:

orthorombic_to_hexagonal_cell.pdf

---

## Round 2 · Referee Report · Anonymous (Referee 2) · 2022-2-7

Strengths

1- Perform state-of-the-art First-Principles calculation of the phonon-assisted luminescence in strained hBN

Weaknesses

1- the reduction of the S3/S1 ratio is too small compared to experiments

Report

Lechifflard et al. addressed the problem of the strain-dependent optical properties in hexagonal boron nitride. They used well established techniques to study the strain-dependence of the band gap and the phonon dispersion. They also performed state-of-the-art calculations of the optical properties, using the technique reported in Ref. [23], and commented in detail the results and their comparison with the experiments.

In order to be accepted, the manuscript should match at least one of the following points:
-Detail a groundbreaking theoretical/experimental/computational discovery;
-Present a breakthrough on a previously-identified and long-standing research stumbling block;
-Open a new pathway in an existing or a new research direction, with clear potential for multipronged follow-up work;
-Provide a novel and synergetic link between different research areas.

Given the lack of a breakthrough in the theory as well as in the results, the submission transcends the criteria of SciPost Physics and meets those of SciPost Physics Core, it should be published in the latter.
  • validity: good
  • significance: good
  • originality: ok
  • clarity: high
  • formatting: excellent
  • grammar: excellent

Author:  Claudio Attaccalite  on 2022-03-11  [id 2283]

(in reply to Report 2 on 2022-02-07)
Category:
objection

They used well established techniques to study the strain-dependence of the band gap and the phonon dispersion. They also performed state-of-the-art calculations of the optical properties, using the technique reported in Ref. [23], and commented in detail the results and their comparison with the experiments.

We respectfully do not agree with the referee's statement. Although the study of strain-dependence in band and phonon dispersion is "well established", the study of phonon-assisted luminescence in strained materials is completely new. As far as we know this is the first work that investigates the role of strain - from an ab-initio perspective - on exciton-phonon coupling and luminescence, and for this reason we think it deserves to be published on SciPost Physics.

Additionally, concerning the comparison with experiments, it is true that we do not find the same quantitative results, however the experimental data should not taken as reference, since they are still unpublished and they could be subject to errors. Actually, this happened several times in the field of BN-based luminescence. For example, a few years ago an experimental group claimed that it was not possible to perform luminescence measurements on single layer hBN due to quenching and other effects, while after a few years clear measurements of this quantity were actually performed. Similarly, the initial high-resolution PL spectra for bulk hBN were wrong because of a missing correction for the UV response of the measuring apparatus, which artificially inverted the intensity ratio between the two main peaks. This was found out and corrected a couple of years later, bringing the experimental spectra in agreement with the theory.

Furthermore, in the present case the experiments also incorporate effects that are not taken into account by the present approach, such as sample curvature and surface effects that neither we nor the experimentalists know how to evaluate for the moment.

We think our results could be a useful reference for experimentalists to verify their setup and measurements. We are aware of the limitations in our calculations which are clearly explained in the text, however we are also confident that in the case of hBN more refined calculations - still out of reach for the ab initio community - would not significantly alter our results. This is due to the peculiarly simple nature of the electronic and excitonic landscapes of this material.

Anonymous on 2022-03-29  [id 2333]

(in reply to Claudio Attaccalite on 2022-03-11 [id 2283])

In their reply, the authors addressed the doubts related to the impact of the work.

Regarding the novelty of the analysis, it is correct to state that the strain dependence of the phonon-assisted luminescence has been never investigated before, although relying on established techniques.
Concerning the comparison with experiments, the possible limitations illustrated in the reply are actually realistic.

Overall, I believe that the work from the authors, although relying on established techniques, posses its own degree of novelty, is very interesting and will inspire further researches from other scientists. If the editor believes that these conditions match the spirit of SciPost Physics, then I am ready to change my previous judgment.

---

## Round 2 · Referee Report · Anonymous (Referee 3) · 2022-2-8

Report

In two recent papers (Refs. 3 and 4), researchers including one of the present authors explored (a) excitonic effects in the electronic excitation spectrum of the bulk layered material hexagonal boron nitride, and (b) the role of electron-phonon coupling in permitting strong contributions to the luminescence spectrum of h-BN from excitonic transitions with non-zero wavevectors q associated with the indirect quasiparticle band gaps. These theoretical investigations used high-quality ab initio computational simulation techniques, in particular (i) GW + Bethe-Salpeter within the well-known Yambo code to reveal the electron-hole excitations including excitonic binding, and (ii) ab initio calculations using density-functional theory of the phonon dispersion and the associated electron-phonon matrix elements, taking account of the non-equilibrium occupation numbers necessary for a proper description of luminescence. The overall picture that emerges is of remarkably strong phonon-assisted transitions in the ultraviolet (5-6 eV), much less attenuated relative to the luminescence from direct transitions (1 eV or so higher) than is usually the case.

The present paper extends these impressive computational techniques to the case of strain applied to h-BN: specifically, uniaxial in-layer strain together with relaxation of the atomic positions in the perpendicular directions, relevant because the effect of strain on luminescence is experimentally accessible, and because slightly strained h-BN is an effective lattice-matched substrate for other hexagonal layered materials, potentially in the context of multi-layer optoelectronic devices. The breaking of symmetry by the strain causes splitting of the excitonic peaks and, interestingly, has qualitatively different effects on the direct and phonon-assisted transitions. Strain also considerably changes the relative strength of the acoustic- and optical-phonon-assisted transitions, as also found experimentally in Ref. 37 (but not yet in quantitative agreement regarding the size of this change).

The state-of-the-art calculations described in Refs. 3 and 4, and now extended to the case of the strained material, provide impressive insight into the unusual strength of phonon-assisted optical transitions, as modified by strain, in h-BN. Clear writing and ingenious analysis of the calculations are very helpful to the reader in understanding how the various results and theories link together. Future refinement of the computational techniques may prove fruitful in fully describing some of the higher-order effects seen experimentally, as the authors suggest. Nevertheless in the meantime there is much here that is robust and interesting, and I recommend publication in SciPost Physics.

Requested changes

No changes necessary.

  • validity: high
  • significance: good
  • originality: good
  • clarity: top
  • formatting: excellent
  • grammar: good

Author:  Claudio Attaccalite  on 2022-03-11  [id 2281]

(in reply to Report 3 on 2022-02-08)
Category:
answer to question

We thank the referee for appreciating our manuscript and suggesting the publication in the present form.

---

## Round 3 · Author Response

Dear Editor,

we thank you for sending us the Referees' comments on the manuscript:

“Excitons under strain: light absorption and emission in strained hexagonal boron nitride”

Following the editor’s suggestion we would like to submit a new version of the present manuscript to
SciPost Physics.
The third referee recommended the publication of the manuscript in the present form in SciPost Physics,
while the first one asked "some adjustments and complementary information".
In the present new version of the manuscript we added all required information and computational
details requested by the first referee.
The second referee recommends the publication in SciPost Core because in his/her opinion
our results constitute a "lack of a breakthrough in the theory" and do not reproduce the experimental data.
We disagree with both these statements.
We think that the comparison with experiment cannot be a criterion to discredit this work,
because in the past several experimental results on PL in BN-bases systems have turned out to be incorrect.
Furthermore, in the experiments there are effects not taken into account
by the present approach such as sample curvature and surface effects that neither we
nor the experimentalists know how to evaluate for the moment.
Regarding the "lack of a breakthrough in the theory" we would like to point out
that although there are many works in the literature on the effect of strain on the electronic structure,
to our knowledge this is the first work that investigates how strain modifies the exciton-phonon coupling from first-principles.
In the new version of the manuscript we underlined this important point,
that we think makes our work worth publishing in SciPost Physics.

All changes are detailed in the answer attached to this letter.

Yours sincerely,
Pierre Lechifflart
Fulvio Paleari
Claudio Attaccalite

---

## Round 3 · List of Changes

1) we improved the abstract, now it explicitly mentions the new method for the luminescence calculation

2) add a reference to this new paper "Vibrational Properties in Highly Strained Hexagonal Boron Nitride Bubbles"
where for the first time, strains comparable with the one studied in our work have been realized

3) Added more computational detail on the optical absorption and
luminescence calculation

4) Corrected and improved discussion on bright and dark excitons

5) Added a new figure with the optical absorption under strain

6) Improved conclusions

---

## Editorial Decision

published